# Genetic Screening of a Hungarian Cohort with Focal Dystonia Identified Several Novel Putative Pathogenic Gene Variants

**DOI:** 10.3390/ijms241310745

**Published:** 2023-06-28

**Authors:** András Salamon, Zsófia Flóra Nagy, Margit Pál, Máté Szabó, Ádám Csősz, László Szpisjak, Gabriella Gárdián, Dénes Zádori, Márta Széll, Péter Klivényi

**Affiliations:** 1Department of Neurology, University of Szeged, 6, Semmelweis Str., H-6725 Szeged, Hungary; salamon.andras@med.u-szeged.hu (A.S.); szabo.mate@med.u-szeged.hu (M.S.); csoszadam95@gmail.com (Á.C.); szpisjak.laszlo@med.u-szeged.hu (L.S.); gardian.gabriella@med.u-szeged.hu (G.G.); zadori.denes@med.u-szeged.hu (D.Z.); 2Department of Medical Genetics, University of Szeged, 4, Somogyi Béla Str., H-6720 Szeged, Hungary; zsofia.flora@gmail.com (Z.F.N.); pal.margit@med.u-szeged.hu (M.P.); szell.marta@med.u-szeged.hu (M.S.); 3Institute of Genomic Medicine and Rare Disorders, Semmelweis University, 78/b, Üllői Str., H-1083 Budapest, Hungary; 4ELKH-SZTE Functional Clinical Genetics Research Group, Eötvös Loránd Research Network, 4, Somogyi Béla Str., H-6720 Szeged, Hungary

**Keywords:** dystonia, genetic, focal, cervical dystonia, blepharospasm

## Abstract

Dystonia is a rare movement disorder which is characterized by sustained or intermittent muscle contractions causing abnormal and often repetitive movements, postures, or both. The two most common forms of adult-onset focal dystonia are cervical dystonia (CD) and benign essential blepharospasm (BSP). A total of 121 patients (CD, 74; BSP, 47) were included in the study. The average age of the patients was 64 years. For the next-generation sequencing (NGS) approach, 30 genes were selected on the basis of a thorough search of the scientific literature. Assessment of 30 CD- and BSP-associated genes from 121 patients revealed a total of 209 different heterozygous variants in 24 genes. Established clinical and genetic validity was determined for nine heterozygous variations (three likely pathogenic and six variants of uncertain significance). Detailed genetic examination is an important part of the work-up for focal dystonia forms. To our knowledge, our investigation is the first such study to be carried out in the Middle-European region.

## 1. Introduction

Dystonia is a rare movement disorder which is characterized by sustained or intermittent muscle contractions causing abnormal (e.g., torsional) and often repetitive movements, postures, or both [1]. This disorder is classified along two axes: clinical characteristics and etiology [1]. The distribution of dystonia according to clinical characteristics is classified as follows: (1) focal—one body site is affected; (2) segmental and multifocal—more than one contiguous or noncontiguous body site is affected; (3) hemidystonia—only one side of the body is affected; (4) generalized—trunk and two or more other body sites are affected [1]. The two most common forms of adult-onset focal dystonia are cervical dystonia (CD) and benign essential blepharospasm (BSP).

Onset of CD usually occurs between 45 and 50 years of age [2]. The risk of dystonia spreading to other parts of the body in this population is very low [3]. In addition to the dystonic posture of the cervical region, dystonic tremor and pain are also present in some patients [4,5,6]. For the proper treatment of CD patients, botulinum neurotoxin (BoNT) therapy (types A and B) is essential [7]. Muscles that play a critical role in the development of abnormal posture and/or movement are selected and injected under ultrasound and/or electromyographical control according to the collum-caput concept [6,8].

BSP affects dominantly, but not exclusively, the orbicular oculi muscles [9]. The disease usually starts between the ages of 50 and 70 [9]. In addition to bilateral, synchronous involuntary muscle contractions, BSP is characterized by eyelid-opening apraxia and increased blink rate [9]. The risk of the disease spreading to other parts of the body is higher compared to other types of focal dystonia [10].

Nonmotor symptoms may also appear as part of the disease (burning or dry feeling in the eyes, photophobia) [9]. Depression and obsessive/compulsive symptoms are relatively common in this population [9]. BoNT type A treatment is considered the gold standard for BSP [11].

The importance of recognizing and treating nonmotor symptoms (e.g., executive dysfunction, anxiety, and depression) is becoming increasingly emphasized for both CD and BSP [4].

The emerging understanding of the etiology of dystonia is very complex. An increasing accumulation of convincing evidence suggests genetic causes; however, the actual genetic cause is often difficult to elucidate. Family history is often negative, which can be partially explained by the reduced penetrance exhibited by the disease [12]. The number of known monogenic forms of dystonia is constantly increasing, even though genetic screening studies performed on these patient cohorts sometimes yield very few relevant results. The average hit rate is 37%; however, genetic diagnosis is more likely for cases with an early age of onset and in combination with the dystonia phenotype. In isolated dystonia forms, the average hit rate is only approximately 4% [13]. To increase the positive genetic diagnostic result rate, some authors recommend the introduction of a scoring system that might help clinicians to prioritize their patients for genetic testing [13]. *CIZ1* and *GNAL* variants are the most often identified genes for BSP, whereas *THAP1*, *CIZ1*, *GNAL*, and *ANO3* mutations are detected most often in genetic screening studies of CD [14,15,16,17,18]. However, growing evidence supports the assumption that the genetic background is a significant contributor to the pathogenesis of these diseases.

The purpose of this clinical study was to perform a genetic analysis of a patient subpopulation with the two most common forms of defined focal dystonia, CD and BSP. To this end, we performed targeted next-generation sequencing (NGS) using genes previously associated with different forms of dystonia as reported in the available scientific literature. The fact that, to the best of our knowledge, no similar study has yet been performed for the Middle-European focal dystonia population, emphasizes the importance of this investigation.

## 2. Results

The assessment of the targeted NGS approach revealed altogether 209 different variants in 24 genes from the 30 dystonia-associated genes. The most frequently affected genes were *SYNE1* (39 different variants for 76 patients), *COL6A3* (29 different variants for 36 patients), *ATM* (22 different variants for 63 patients), and *CACNA1B* (20 different variants for 114 patients) (Figure 1). No variants were identified for the following genes: *FTL*, *PANK2*, *RAB12*, *THAP1*, *TUBB4A*, and *WDR45*.

According to the ACMG guidelines, two variants identified in the *ATM* and the *ATP7B* genes are considered pathogenic (Table 1). Four variants are categorized as likely pathogenic: one in *COL6A3* and three in the *GNAL* genes (Table 1). Fifty-three variants are considered variant of uncertain significance (VUS) with pathogenic favor (Appendix A).

The two pathogenic variants and one of the likely pathogenic variants (*COL6A3*: p.Gln2245Ter) are unlikely to be disease-causing for the affected patients that carry them as they are autosomal recessive, and the identified variants occur in the heterozygous state. Obviously, it is also possible that the second variant required for autosomal recessive inheritance was not identified in our study. Intronic, including deep intronic, mutations can create strong cryptic splice acceptor sites leading to frameshift mutations; however, they cannot be identified by NGS panels and WES.

All 53 VUS were reviewed in association with patient symptoms to validate their clinical significance. From the identified 53 VUS, six variants can be considered as clinically relevant in our cohort: one in each of the *ANO3*, *CIZ1*, *KCNN2*, and *KMT2B* genes and two in the *VPS16* gene (Table 1). All of these variants were identified in heterozygous form, which is in agreement with the reported mode of inheritance for these genes. The majority of the identified variants are missense, complicating the consideration of causality. Only one nonsense and three splice region variants were identified.

From the 14 cases subjected to WES, genetic diagnosis was established for two cases in which a causative likely pathogenic *GNAL* variant and the previously mentioned *KCNN2* VUS were identified. No differences were found during the CNV analyses.

## 3. Discussion

In the present clinical study of focal dystonia, we performed targeted NGS examination. For 114 cases, gene panels (with 30 genes) were used for testing. Whole-exome sequencing (WES) was performed for 14 additional cases (seven of these patients were previously tested using gene panels).

In accordance with the scientific literature, we detected potentially relevant variations in known genes previously associated with CD and BSP, such as *CIZ1*, *GNAL*, and *ANO3*, and in other dystonia-related genes, such as *KMT2B*, *VPS16*, and *KCNN2*, for a total of nine patients (Table 1).

In the case of *GNAL* gene, three likely pathogenic variants are considered significant.

The *GNAL* c.677G>T (p.Cys226Phe) variant was identified from a male patient who suffers from left-sided neck pain and right-directed laterocollis, torticollis, and mild retrocollis, as well as periodic “yes–yes” dystonic tremor, since the age of 27. The currently 49 year old unaffected mother of the patient also carries this variant in a heterozygous form. In our opinion, the fact that the mother is unaffected is likely due to the low penetrance characteristic of dystonia.

The symptoms of the female patient with the second *GNAL* c.1315G>A (p.Val439Met) variant started at the age of 30. The complaint of the patient was that her head started to turn slightly to the left and it was increasingly difficult for her to keep it in the middle position. During physical examination, we noticed a torticollis to the left, as well as pain on the left side of the trapezius muscle. No other member of the family was affected, and none of the family members wished to participate in genetic testing.

The *GNAL* c.1288G>A (p.Ala430Thr) variant was identified for a female patient with CD (left-directed laterocollis, torticollis, and antecollis with platysmal contractions) since the age of 19. She received BoNT and deep-brain stimulation (bilateral globus pallidus internus) treatments, but they were not effective. The complaints of the patient show a progressive nature. The mother and a sister of the patient have similar symptoms but were not able to participate in genetic testing.

*GNAL* variants have been identified in less than 2% of dystonia patients of European origin [20]. In our study, we report a higher frequency compared to previous reports. All three variants described here are considered likely pathogenic, and the clinical phenotype matches the features previously associated with *GNAL* variants.

For rare genetic diseases, familial segregation analysis provides valuable information about the pathogenicity of the identified variant. In the case of dystonia, the variable onset of the symptoms and the incomplete penetrance of the disease make analysis especially difficult. Analysis is further complicated by the missense nature of the variants. Following careful consideration based on the symptoms of patients, six VUS (five identified with gene panels and one with WES) were considered clinically relevant: one in each of the *ANO3*, *CIZ1*, *KCNN2*, and *KMT2B* genes and two in the *VPS16* gene (Table 1).

One male patient with the c.2276-6T>C *ANO3* splice region variant has had CD (left-sided torticollis, right-sided laterocaput, and mild retrocollis) and dysarthria since the age of 41. No family members were available for family testing.

A novel VUS, c.1820A>G (p.Glu607Gly) in the *CIZ1* gene, was identified from a male CD patient whose symptoms of right-directed torticollis and dysphonia started at the age of 20. Unfortunately, no family members were available for segregation analysis of this variant.

The *KCNN2* c.1625G>A (p.Arg542Gln) variant identified by WES affects a highly conserved amino-acid position. The affected patient has had involuntary twitching of the head since the age of 21. We found the following alterations during the neurological examination: CD symptoms include right torticollis, slight laterocaput to the left, “no–no” tremor, and left sternocleidomastoid and the trapezius bundled on both sides. Mild cerebellar symptoms and slight postural tremor were also present. We did not observe any myoclonic jerk or significant eye movement abnormality. The brain MRI showed a subcortical vascular malformation in the right hemisphere. The unaffected mother of the proband does not carry the variant.

The *KMT2B* c.3136C>T (p.Arg1046Cys) variant was discovered in a CD patient. The symptoms of the female patient started when she was 34 years old (left-sided laterocollis, torticollis, and antecollis). BoNT was only partially effective. Dystonic symptoms improved significantly, in agreement with the literature, after bilateral globus pallidus internus deep-brain stimulation. None of the asymptomatic family members carried the variant, which supports the possibility of the p.Arg1046Cys VUS being disease-causing.

Two rare variants of uncertain significance were uncovered in the *VPS16* gene. The c.1370T>C (p.Leu457Pro) missense variant was identified in a male CD patient with retrocollis and dysarthria, and the c.241-2A>C splice site variant was found in a female with BSP. *VPS16* variants were first described in the background of an early-onset generalized dystonic disorder with lysosomal dysfunction [21]. However, *VPS16* variants have recently been detected in patients with focal dystonia [22]. Our findings further suggest that *VPS16* variants may be considered in cases of focal dystonia.

We believe that variants we identified in genes not yet associated with CD or BSP may extend the phenotypic spectrum of these genes.

In summary, a genetically supported diagnosis was possible for nine cases of the 121 patients included in the study (hit rate: 7.4%), and this rate is in good agreement with the literature (4%) [13]. The WES examination performed for 14 cases revealed only a likely pathogenic variant and a VUS in the *GNAL* and *KCNN2* genes in two cases. In our cohort, we found no differences in other frequently reported genes (*AOPEP* and *EIF2AK2*) [23]. Except for one case (*VPS16*), all variants considered relevant were found in patients with CD. The average age of patients was 64 years in our cohort; however, in agreement with the literature, we found relevant variants almost exclusively in patients with earlier disease onset (mean: 31.6 years) [24]. A limitation of our study is that no functional testing was performed for the variants found, and the segregation analysis was not informative in the vast majority of cases.

In conclusion, with our study—which we believe is the only thorough study of a relatively large Middle-European focal dystonia population—we were able to further strengthen the potential pathogenic role of novel variants in dystonia-related genes. Furthermore, we raised the possibility of potentially novel/different phenotypes of known disease-associated genes. Although we identified several variants, functional studies with these variants (e.g., PAXgene-based or minigene assay), as well as the detection of additional variants in this patient group, are necessary to strengthen their pathogenic role. As more and more information and evidence are becoming available regarding genes involved in dystonia, it is likely that the status of our VUSs may change to pathogenic, and that our diagnostic yield will increase. Given that WES analysis is not expected to have a significantly higher hit rate compared to the use of gene panels, we recommend the use of the panel we compiled for CD patients, particularly for cases when budget is limited. Nevertheless, it is important to mention that the identification of a monogenic etiology in the background of dystonia is expected to be rare, as other nongenetic etiological factors, including developmental issues, environmental influences, epigenetic modifications, or polygenic interactions, may also influence the development of the disorder. However, if there is reasonable suspicion that the form of dystonia is genetically determined (e.g., positive family history; onset of the disease before the age of 20; other co-existing (neurological and non-neurological) symptoms), NGS may provide an exceptional opportunity to confirm the disease-causing element [13]. In our opinion, in the near future, long-read sequencing analysis can be expected to gain popularity in the field of neurological diseases, including dystonia [25].

We hope that the information provided by the current study will aid the clinical research for these genetically heterogeneous disorders. In terms of the applicability in clinical environments, we see great benefit from performing detailed genetic examination for generalized forms of dystonia and those combined with other movement disorders, as well as for focal dystonia forms. We anticipate that the results of such genetic analyses will be advantageous in the development of personalized treatment options.

## 4. Patients and Methods

### 4.1. Patients

Overall, 74 CD and 47 BSP nonrelated patients without a previous genetic diagnosis were enrolled in the study. The gender distribution was as follows: 31 men and 90 women. The average age of the patients was 64 years (range: 24–94 years). Patient inclusion in the study was decided under the supervision of movement disorder experts at the Department of Neurology of the University of Szeged. All patients fulfilled the current diagnostic criteria for the corresponding focal dystonia form [1,26,27]. Written informed consent was obtained from all patients participating in this study (Regional Human Biomedical Research Ethics Committee of the University of Szeged registration number is 22/2021). All procedures performed in this study involving human participants were in accordance with the ethical standards of the Regional Human Biomedical Research Ethics Committee of the University of Szeged and with the 1964 Helsinki Declaration and its later amendments or comparable ethical standards.

### 4.2. Methods

For genomic DNA extraction, anticoagulated venous blood was collected, and DNA extraction was performed according to the manufacturer’s protocol (QIAGEN GmbH, Hilden, Germany). For the screening of genes associated with CD or BSP, a targeted NGS approach was used for all 121 patients. For 114 cases, gene panels were used for testing. Whole-exome sequencing (WES) was performed for 14 additional cases (seven of these patients were previously tested using gene panels). Following a thorough search of the scientific literature, we focused strongly on 30 genes previously associated with dystonia (Table 2). RNA probes for the selected loci were designed with SureDesign software (19 July 2009, Agilent Technologies, Santa Clara, CA, USA), and the SureSelect^QXT^ Reagent Kit (Agilent Technologies, Santa Clara, CA, USA) was used for the preparation of libraries. NGS was carried out on an Illumina NextSeq 550 NGS platform using the 300-cycle Mid Output Kit v2.5 (Illumina, Inc., San Diego, CA, USA). Paired-end reads were aligned against the hg19 Human Reference Genome with the Burrows–Wheeler Aligner. Genome Analysis Toolkit was used for variant-calling purposes. With this technique, a mean on-target coverage of 290.9 was achieved, and 97.8% of the targets were covered at least 20×. For the annotation of variants, the ANNOVAR software tool was used (version 2.0.2, 17 July 2017) [21]. Variant interpretation was carried out in accordance with the 2015 guidelines of the American College of Medical Genetics and Genomics [28] using Franklin bioinformatic sites (https://franklin.genoox.com accessed on 13 January 2023). We also applied VarSome [29] and SpliceAI [30]. Variants of interest were validated with bidirectional Sanger sequencing. In the 14 cases in which WES was performed, copy number variation (CNV) analysis was also completed.

## Figures and Tables

**Figure 1 ijms-24-10745-f001:**
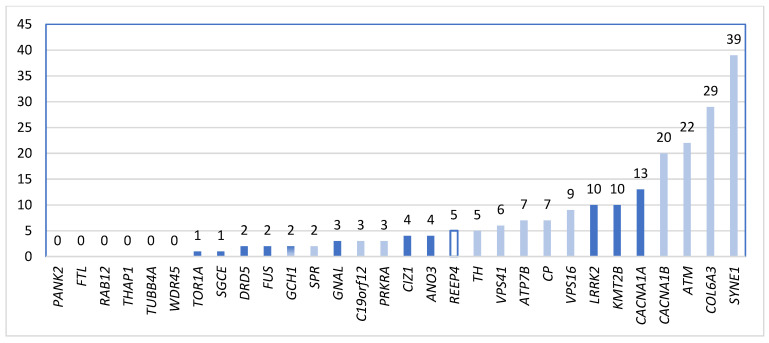
The distribution of the variants for each investigated gene (all patients; n = 121). The autosomal dominant inheritance mode is indicated with dark color, whereas the recessive is indicated with light color. No inheritance data are available for *REEP4* (transparent color). Interestingly, the most frequently identified variation in our population was in the *SYNE1* gene (Synofzik et al. [19] have already shown that *SYNE1* mutations can not only cause pure cerebellar ataxias). Abbreviations: *ANO3*—anoctamin 3; *ATM*—ataxia-telangiectasia mutated; *ATP7B*—ATPase copper transporting beta; *C19orf12*—chromosome 19 open reading frame 12; *CACNA1A*—calcium channel, voltage-dependent, P/Q type, alpha-1A subunit; *CACNA1B*—calcium channel, voltage-dependent, N type, alpha-1B subunit; *CIZ1*—CIP1-interacting zinc finger protein; *COL6A3*—collagen, type VI, alpha-3; *CP*—ceruloplasmin; *DRD5*—dopamine receptor D5; *FTL*—ferritin light chain; *FUS*—fused in sarcoma; *GCH1*—GTP cyclohydrolase 1; *GNAL*—guanine nucleotide-binding protein alpha-activating activity polypeptide, olfactory type; *KMT2B*—lysine-specific methyltransferase 2B; *LRRK2*—leucine-rich repeat kinase 2; *PANK2*—pantothenate kinase 2; *PRKRA*—protein kinase, interferon-inducible double-stranded RNA-dependent activator; *RAB12*—RAS-associated protein RAB12; *REEP4*—receptor expression-enhancing protein 4; *SGCE*—sarcoglycan, epsilon; *SPR*—sepiapterin reductase; *SYNE1*—spectrin repeat-containing nuclear envelope protein 1; *TH*—tyrosine hydroxylase; *THAP1*—THAP domain-containing protein 1; *TOR1A*—torsin 1A; *TUBB4A*—tubulin, beta-4A; *VPS16*—VPS16 core subunit of corvet and HOPS complexes; *VPS41*—VPS41 subunit of HOPS complex; *WDR45*—WD repeat-containing protein 45.

**Table 1 ijms-24-10745-t001:** The main genetic findings of the study.

Gene (Inheritance)	Transcript Number	cDNA Position	Protein Position	Zygosity	MAF	ACMG Classification	PhyloP100 Way Score	Focal Dystonia Type	Detection Method
*ATM* (AR)	NM_000051.3	c.6095G>A	p.Arg2032Lys	Heterozygous	0.0062%	Pathogenic (PM2, PP3, PP5)	9.349	CD	PS
*ATP7B* (AR)	NM_000053.4	c.2605G>A	p.Gly869Arg	Heterozygous	0.1251%	Pathogenic (PM1, PP2, PM2, PM5, PP3, PP5)	7.735	CD	PS
*COL6A3* (AR)	NM_004369.4	c.6733C>T	p.Gln2245Ter	Heterozygous	0	Likely pathogenic (PVS1, PM2)	5.899	BSP	PS
*GNAL* (AD)	NM_182978.4	c.677G>T	p.Cys226Phe	Heterozygous	0	Likely pathogenic (PP3, PM2, PP2)	8.978	CD	PS
*GNAL* (AD)	NM_182978.4	c.1315G>A	p.Val439Met	Heterozygous	0	Likely pathogenic (PP3, PM2, PP2)	9.509	CD	PS
*GNAL* (AD)	NM_182978.4	c.1288G>A	p.Ala430Thr	Heterozygous	0	Likely pathogenic (PP3, PM2, PP2)	9.781	CD	WES
*ANO3* (AD)	NM_031418.4	c.2276-6T>C	-	Heterozygous	0.0062%	VUS (PM2)	−0.004	CD	PS
*CIZ1* (AD)	NM_001131016.2	c.1820A>G	p.Glu607Gly	Heterozygous	0	VUS (PM2, BP4)	4.452	CD	PS
*KCNN2* (AD)	NM_021614.4	c.1625G>A	p.Arg542Gln	Heterozygous	0.0009%	VUS (PM2, PP2)	9.755	CD	WES
*KMT2B* (AD)	NM_014727.2	c.3136C>T	p.Arg1046Cys	Heterozygous	0	VUS (PM2, PP2)	3.599	CD	PS
*VPS16* (AD, AR)	NM_022575.4	c.1370T>C	p.Leu457Pro	Heterozygous	0	VUS (PM2, PP3)	7.103	CD	PS
*VPS16* (AD, AR)	NM_022575.4	c.241-2A>C	-	Heterozygous	0	VUS (PM2, PVS1)	6.559	BSP	PS

Abbreviations: AD—autosomal dominant; AR—autosomal recessive; BSP—benign essential blepharospasm; CD—cervical dystonia; MAF—minor allele frequency in the gnomAD database among non-Finnish Europeans; PS—panel sequencing; WES—whole-exome sequencing. Phylo P score is an indicator of evolutionary conservation at each site compared to a hypothetical neutral drift. The higher the positive score, the more conserved the site is, while negative scores indicate an acceleration in variation.

**Table 2 ijms-24-10745-t002:** List of 30 genes examined in our focal dystonia population.

*ANO3*	*CACNA1B*	*FTL*	*LRRK2*	*SGCE*	*TOR1A*
*ATM*	*CIZ1*	*FUS*	*PANK2*	*SPR*	*TUBB4A*
*ATP7B*	*COL6A3*	*GCH1*	*PRKRA*	*SYNE1*	*VPS16*
*C19orf12*	*CP*	*GNAL*	*RAB12*	*TH*	*VPS41*
*CACNA1A*	*DRD5*	*KMT2B*	*REEP4*	*THAP1*	*WDR45*

Abbreviations: *ANO3*—anoctamin 3 [31]; *ATM*—ataxia-telangiectasia mutated [32]; *ATP7B*—ATPase copper transporting beta [33]; *C19orf12*—chromosome 19 open reading frame 12 [34]; *CACNA1A*—calcium channel, voltage-dependent, P/Q type, alpha-1A subunit [35]; *CACNA1B*—calcium channel, voltage-dependent, N type, alpha-1B subunit [36]; *CIZ1*—CIP1-interacting zinc finger protein [37]; *COL6A3*—collagen, type VI, alpha-3 [38]; *CP*—ceruloplasmin [39]; *DRD5*—dopamine receptor D5 [40]; *FTL*—ferritin light chain [41]; *FUS*—fused in sarcoma [15]; *GCH1*—GTP cyclohydrolase 1 [42]; *GNAL*—guanine nucleotide-binding protein alpha-activating activity polypeptide, olfactory type [20]; *KMT2B*—lysine-specific methyltransferase 2B [43]; *LRRK2*—leucine-rich repeat kinase 2 [15]; *PANK2*—pantothenate kinase 2 [44]; *PRKRA*—protein kinase, interferon-inducible double-stranded RNA-dependent activator [17]; *RAB12*—RAS-associated protein RAB12 [45]; *REEP4*—receptor expression-enhancing protein 4 [16]; *SGCE*—sarcoglycan, epsilon [46]; *SPR*—sepiapterin reductase [47]; *SYNE1*—spectrin repeat-containing nuclear envelope protein 1 [15]; *TH*—tyrosine hydroxylase [48]; *THAP1*—THAP domain-containing protein 1 [49]; *TOR1A*—torsin 1A [50]; *TUBB4A*—tubulin, beta-4A [51]; *VPS16*—VPS16 core subunit of corvet and HOPS complexes [52]; *VPS41*—VPS41 subunit of HOPS complex [21]; *WDR45*—WD repeat-containing protein 45 [53].

## Data Availability

Data is available from the authors upon request.

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
