# Peer review of "Genetic Screening of a Hungarian Cohort with Focal Dystonia Identified Several Novel Putative Pathogenic Gene Variants"

_ijms, 2023, doi:10.3390/ijms241310745_

Round 1

Reviewer 1 Report

The authors performed a genetic screening of a cohort of Hungarian patients affected by cervical dystonia (CD) or benign essential blepharospasm (BSP) using a NGS custom panel of 30 dystonia-associated genes (114 patients) or WES techniques (14 patients). They identified 209 variants in 25 genes and tried to establish whether one of these was the causative variant of the disease. Contrary to what the authors claim, for none of the subjects it was possible to establish the causative gene because either the other family members were not available or they were not informative. Not functional studies were performed. Currently more than 90 causative genes of dystonia have been described, the choice of the authors on the 30 genes present in the panel is not clear given that some of these were never even associated with CD or BSP

Both the text and the figures are full of inaccuracies. Some examples:

-   Line 110: the author write 54 variants but the variants are 53 as written correctly in the line 133

-        Line 160: The author described the “third” GNAL variant and the second was reported after (line 166)

-        Line 152: is Table 1 not Table 2 as written

-        Table 2: The gene list is repeated twice

Author Response

We thank Reviewer 1 for the careful reading and valuable comments, we revised the manuscript accordingly.

Question 1: The authors performed a genetic screening of a cohort of Hungarian patients affected by cervical dystonia (CD) or benign essential blepharospasm (BSP) using a NGS custom panel of 30 dystonia-associated genes (114 patients) or WES techniques (14 patients). They identified 209 variants in 25 genes and tried to establish whether one of these was the causative variant of the disease. Contrary to what the authors claim, for none of the subjects it was possible to establish the causative gene because either the other family members were not available or they were not informative. Not functional studies were performed. Currently more than 90 causative genes of dystonia have been described, the choice of the authors on the 30 genes present in the panel is not clear given that some of these were never even associated with CD or BSP.

Response 1: The genetic variations identified in our study were classified and indicated according to the currently valid ACMG classification (Richards et al., 2015, doi: 10.1038/gim.2015.30) (see Table 1). Nevertheless, the Reviewer is right in that the text may have unintentionally placed increased emphasis on the pathological role of individual genetic variations. We modified the text accordingly. The Reviewer is also right in that the number of genes underlying dystonia is rapidly increasing. However, not all of these genes are associated with focal forms of dystonia. References to the gene elements of the dystonia panel used in our study are indicated in Table 2 (abbreviations). When compiling the panel, we tried to include in the list genes for which only limited literature data are available in relation to focal dystonia.

Question 2: Both the text and the figures are full of inaccuracies. Some examples: Line 110: the author write 54 variants, but the variants are 53 as written correctly in the line 133. Line 160: The author described the “third” GNAL variant and the second was reported after (line 166). Line 152: is Table 1 not Table 2 as written. Table 2: The gene list is repeated twice.

Response 2: Thank you for your comment. Inaccuracies in the text have been corrected.

Reviewer 2 Report

The authors included 121 patients (cervical dystonia - CD, 74; blepharospasm - BSP, 47) in the 20 study. Thirty genes were selected based on a thorough search of the scientific literature. They identified a total of 209 different heterozygous variants in 24 out of the 30 genes. The article is well-framed and reads very well. I do have some minor comments:

1) In the definition of dystonia, please include that movements are "torsional".

2) Regarding dystonia distribution, I would specify that it involve one or multiple muscular groups (instead of body sites) and that in case of hemi dystonia it involves the whole embody (and would avoid saying "only").

3) I would add a comment on the results of the "SYNE1” gene. SYNE1 is one of the largest genes in human brain, this is why the clinical features may range from pure cerebeellar ataxia to ataxia plus several other neurological and non-neurological symptoms. See also: Synofzik M, et al. SYNE1 ataxia is a common recessive ataxia with major non-cerebellar features: a large multi-centre study. Brain. 2016 May;139(Pt 5):1378-93. doi: 10.1093/brain/aww079.

4)  In the discussion, I would ask the authors to propose a sort of algorithm or flow-chart, about doing genetic panels in all patients (as they currently state), and then repeating WES only in those negative and with early age at onset or where a genetic nature of the disease is strongly suspected (e.g., family history). Then, in those with negative results, what about introducing targeted long-read sequencing analysis? It is known that many repeat-expansion diseases may manifest with tremor and dystonic phenotypes (e.g., NOTCH2NLC). See also: Marsili L, et al. Uncovering Essential Tremor Genetics: The Promise of Long-Read Sequencing. Front Neurol. 2022 Mar 23;13:821189. doi: 10.3389/fneur.2022.821189. 

Minor comments:

-       Please, avoid saying “VUS variant”, is redundant. You may want to mention only “VUS”.

-       Table 2 format is weird, please, correct it avoiding dashed lines.

English language needs minor editing 

Author Response

We thank the Reviewer for the careful reading and valuable comments, we revised the manuscript accordingly.

The authors included 121 patients (cervical dystonia - CD, 74; blepharospasm - BSP, 47) in the 20 study. Thirty genes were selected based on a thorough search of the scientific literature. They identified a total of 209 different heterozygous variants in 24 out of the 30 genes. The article is well-framed and reads very well.

I do have some minor comments:

Question 1: In the definition of dystonia, please include that movements are "torsional".

Response 1: Although the term "torsional" is not included in the currently used definition of dystonia (Albanese et al., 2014, doi: 10.1002/mds.25475), however the Reviewer is right that torsional features may also be present in the clinical picture of dystonia, so we have indicated this in the text.

Question 2: Regarding dystonia distribution, I would specify that it involve one or multiple muscular groups (instead of body sites) and that in case of hemi dystonia it involves the whole embody (and would avoid saying "only").

Response 2: During our clinical survay, we focused exclusively on certain focal forms of dystonia, i.e., cervical dystonia and benign essential blepharospasm. Patients with multifocal, segmental, hemi- or generalized dystonia were not included into our study. In a previous work, we performed a detailed analysis of our population with cervical dystonia regarding the involved muscles in comparison with international data (Szabó et al., 2023, doi: 10.18071/isz.76.0037). However, the purpose of the current study was not the identification of muscles involved in the development of certain forms of focal dystonia (CD and BSP)  but the genetic screening of our patients.

Question 3: I would add a comment on the results of the "SYNE1” gene. SYNE1 is one of the largest genes in human brain, this is why the clinical features may range from pure cerebellar ataxia to ataxia plus several other neurological and non-neurological symptoms. See also: Synofzik M, et al. SYNE1 ataxia is a common recessive ataxia with major non-cerebellar features: a large multi-centre study. Brain. 2016 May;139(Pt 5):1378-93. doi: 10.1093/brain/aww079.

Response 3: Thank you for your comment. We supplemented the manuscript with the above important work.

Question 4: In the discussion, I would ask the authors to propose a sort of algorithm or flow-chart, about doing genetic panels in all patients (as they currently state), and then repeating WES only in those negative and with early age at onset or where a genetic nature of the disease is strongly suspected (e.g., family history). Then, in those with negative results, what about introducing targeted long-read sequencing analysis? It is known that many repeat-expansion diseases may manifest with tremor and dystonic phenotypes (e.g., NOTCH2NLC). See also: Marsili L, et al. Uncovering Essential Tremor Genetics: The Promise of Long-Read Sequencing. Front Neurol. 2022 Mar 23;13:821189. doi: 10.3389/fneur.2022.821189. 

Response 4:  Zech et al. (2020, doi: 10.1016/S1474-4422(20)30312-4) formulated a proposal for the genetic testing of patients with dystonia. In the light of our own results, the Discussion section has been supplemented with this critically important aspect. Although long-read sequencing analysis is not part of the widely used genetic investigation of dystonia, however the Reviewer is right that this methodology can be an additional element in the future genetic investigation of dystonia. We supplemented the text with this information.

Question 5: Please, avoid saying “VUS variant”, is redundant. You may want to mention only “VUS”.

Response 5: Thank you for the comment. We have made the appropriate changes in the text.

Question 6: Table 2 format is weird, please, correct it avoiding dashed lines.

Response 6: - Table 2 has been reformatted.

Question 7: English language needs minor editing.

Response 7:  The submitted version of the manuscript was reviewed by a native English speaker with PhD in biology and decades of expertise in language editing.

Round 2

Reviewer 1 Report

The authors answered to all my comments